# Quartet: Native FP4 Training Can Be Optimal for Large Language Models

**Roberto L. Castro**[*]
ISTA & Red Hat AI

**Andrei Panferov**[*]
ISTA

**Soroush Tabesh**
ISTA

**Oliver Sieberling**
ETH Zürich

**Jiale Chen**
ISTA

**Mahdi Nikdan**
ISTA

**Saleh Ashkboos**
ETH Zürich

**Dan Alistarh**
ISTA & Red Hat AI

## Abstract

Training large language models (LLMs) models directly in low-precision offers a way to address computational costs by improving both throughput and energy efficiency. For those purposes, NVIDIA's recent Blackwell architecture facilitates very low-precision operations using FP4 variants. Yet, current algorithms for training LLMs in FP4 precision face significant accuracy degradation and often rely on mixed-precision fallbacks. In this paper, we investigate hardware-supported FP4 training and introduce a new approach for accurate, end-to-end FP4 training with all the major computations (i.e., linear layers) in low precision. Through extensive evaluations on Llama-type models, we reveal a new low-precision scaling law that quantifies performance trade-offs across bit-widths and training setups. Guided by this investigation, we design an "optimal" technique in terms of accuracy-vs-computation, called Quartet. We implement Quartet using optimized CUDA kernels tailored for Blackwell, demonstrating that fully FP4-based training is a competitive alternative to FP16 half-precision and to FP8 training. Our code is available at `https://github.com/IST-DASLab/Quartet`.

## 1 Introduction

Over the past decade, the capabilities of large language models (LLMs) have surged, unlocking state-of-the-art performance in AI reasoning, coding, and multimodal understanding. These advances have come at the cost of an unprecedented rise in compute costs, as the floating-point operations (FLOPs) required to train a frontier model have been doubling every few months [16].

One key lever for reducing compute costs is *lower-precision computation*: executing the matrix-multiplication (MatMul) kernels that dominate training workloads at lower bit-widths yields near-linear gains in throughput and energy efficiency. On the inference side, it is known that 4-bit quantization—or even lower—can preserve accuracy, via sophisticated calibration and rotation schemes [24; 2; 9]. For training, recent work has pushed the precision frontier from FP16 [33] to 8-bit pipelines, responsible in part for efficiency breakthroughs such as DeepSeek-V3 [31]. In this context, NVIDIA's Blackwell architecture introduces efficient hardware support for even lower-precision microscaling formats [36] such as MXFP and NVFP, which natively support 4-bit floating-point operations at higher teraFLOP-per-watt efficiency: for instance, moving from 8- to 4-bit multiplies on the B200 GPU can almost *double* arithmetic throughput, while cutting energy roughly in half [35].

Yet, today's algorithmic support for *accurate end-to-end* training in such low precision is missing. State-of-the-art quantized training methods such as Switchback [54], Jetfire [57], HALO [3], and

---

[*]- Equal contribution. Correspondence to: `dan.alistarh@ist.ac.at`.

39th Conference on Neural Information Processing Systems (NeurIPS 2025).

INT4-Transformers [56] either (i) lose precision and stability when training current models in 4-bit formats, or (ii) fall back to higher precision for selected matrix multiplications. Bridging this gap calls for both a deeper understanding of quantization error during back-propagation and new algorithmic safeguards tailored to hardware-native FP4 formats.

**Contributions.** In this paper, we address this challenge via a first systematic study of hardware-supported FP4 training, focusing on the high-efficiency of the MXFP4 format [36; 35]. Based on this analysis, we introduce an algorithm for MXFP4 native training—in which all matrix multiplications occur in MXFP4—called **Quartet**, which provides the best accuracy-efficiency trade-off among existing methods, and is near-lossless for LLM pre-training in the large-data regime. Our main technical contribution is a highly-efficient GPU implementation of Quartet, which achieves speedups of almost 2x relative to FP8 for linear layer computations on an NVIDIA Blackwell RTX 5090 GPU. One key achievement is that Quartet enables MXFP4 precision to be "optimal" on the accuracy-efficiency trade-off: at a fixed computational budget, the accuracy impact of lower-precision training in Quartet is fully compensated by the higher efficiency of our implementation. In more detail, our contributions are as follows:

1. We propose and implement a new approach for comparing quantized training methods, via *their induced scaling law*, which dictates the loss achievable under a specific computation and data budget. We propose and fit such a law for all existing methods, isolating two key parameters: the *parameter efficiency* $\text{eff}_N$ of each method, and its *data efficiency* $\text{eff}_D$. A method is superior to another if it improves upon both these metrics.

2. We find that the *parameter efficiency* is directly linked to the *forward compression error* of each training method, whereas *data efficiency* is linked to the bias in the method's gradient estimator, which we measure via a novel *misalignment* metric. Given a computational and data budget, and real-world speedups due to lower precision, these metrics allow us to predict the "optimal" low-precision setup to train a given model to a target accuracy, maximizing accuracy-vs-runtime.

3. We apply this framework to MXFP4 precision, seeking to determine if there are practical settings under which native training in this precision can be optimal on Blackwell GPUs. We isolate an algorithm, called Quartet, which achieves this by maximizing both parameter and data efficiency, building on previous SOTA methods for QAT [37] and quantized backward-pass optimization [47]. Our key technical contribution is a complex, highly-efficient GPU implementation of Quartet specialized to the new Blackwell architecture.

4. We validate our approach experimentally by pre-training Llama-family [46] models on the C4 dataset [39]. Our experiments show that 1) Quartet provides superior accuracy relative to prior methods [56; 58; 3] across different computing budgets and model sizes, and that 2) its fast implementation allows it to outperform highly-optimized FP8 kernels. This establishes that MXFP4 can indeed provide "optimal" training in practice.

Our work bridges the gap between emerging low-precision hardware capabilities and the algorithmic support needed for accurate, end-to-end quantized model training. Specifically, we show for the first time that the new MXFP4 format can be competitive with FP8 in terms of accuracy-vs-speed, which we hope can enable significant reductions in the rising computational costs of AI.

## 2 Related Work

**Training in 8-bit formats.** Early work on low-precision neural network training focused on 8-bit or higher precisions, mainly on CNNs. Banner et al. [4] demonstrated accurate 8-bit training via careful scaling and higher-precision accumulation. Yang et al. [59] proposed a framework that quantized weights, activations, gradients, errors, and even optimizer states to INT, achieving for the first time completely integer-only training with comparable accuracy. SwitchBack [55] and JetFire [58] build on this progress, targeting 8-bit training for Transformers [50]. Specifically, SwitchBack uses a hybrid INT8/BF16 linear layer for vision-language models, performing forward and input-gradient MatMuls in INT8 while computing weight gradients in 16-bit; this yielded 13–25% end-to-end speedups on CLIP models with accuracy within 0.1% of full precision.

JetFire [58] achieved *fully* INT8 training for Transformers by using a novel per-block quantization scheme to handle activation and gradient outliers. By partitioning matrices into small blocks and scaling each block independently, JetFire preserved accuracy comparable to FP16 training while

obtaining $\sim 40\%$ end-to-end speedup and $1.49\times$ reduction in memory usage. The JetFire approach is conceptually similar to the FP8 DeepSeek training technique [31], which used larger block sizes. Recently, HALO [3] improved upon JetFire in terms of the accuracy-speedup trade-off in INT8, specifically focusing on low-precision fine-tuning. **In our work, we will treat FP8 as the idealized baseline that has the quality of BF16 and the speed of raw FP8 GEMM operations.** That is, when comparing agains FP8, we compare against simultaneously the most accurate and the fastest FP8-based methods could ever be.

**End-to-end lower-precision training.** As our results and prior work suggest, going below 8-bit precision in training using the above approaches is extremely challenging, due to the narrower dynamic range and higher error. This frontier was first explored by Sun et al. [42], who achieved 4-bit training on ResNets by using a custom numeric format, which unfortunately is far from being supported in hardware. Chmiel et al. [11] introduced a logarithmic unbiased quantization (LUQ) scheme to this end, combining two prior ideas: (1) a log-scale FP4-type format to cover a wider dynamic range, and (2) applying stochastic unbiased rounding on the backward. For reference, LUQ incurs a $1.1\%$ top-1 accuracy drop on ResNet50/ImageNet, and has not been validated on hardware-supported FP formats. Xi et al. [56] proposed a method to train Transformers using INT4 effective precision for all linear layers, using specialized quantizers: block-wise Hadamard transform and LSQ [20] for outlier mitigation on the forward pass, and leverage score sampling on the backward pass to exploit structured sparsity, together with a custom INT4-effective format. Their approach trains BERT-family models within 1-2% accuracy gap relative to FP16, with a 2.2x speedup on individual matrix multiplies (relative to 4x theoretical speedup), leading to up to 35% faster training end-to-end.

We compare relative to these techniques in Section 5, and show that Quartet outperforms them significantly in terms of accuracy and stability.

**Mixed-precision training in low-precision formats.** Given the importance of inference cost reductions, there has been significant work on *quantization-aware training (QAT)* [14; 7; 20; 5; 52; 29], i.e., methods that only quantize the *forward pass*. Two key difficulties in this setting are 1) minimizing the error induced by quantization on the forward pass, and 2) obtaining a stable gradient estimator over the resulting discrete space. With regards to error reduction, existing methods either try to find a good "learnable" fit w.r.t. the underlying continuous distribution [14; 20], or perform noise injection during QAT in order to make the network more robust to quantization [5]. Closer to our work, Wang et al. [53] explored FP4 QAT, introducing a "smoother" gradient estimator, together with outlier clamping and compensation to handle activation outliers. While their approach shows good accuracy, it is fairly complex and not validated in terms of efficient support. Prior work by [37] provided a simpler alternative approach, based on more precise MSE fitting, an optional Hadamard rotation, and a clipping-aware "trust" gradient estimator. By contrast with these forward-only approaches, recent work by Tseng et al. [47] investigated *backward-only* quantization with the MXFP4 format, signaling the importance of stochastic rounding and outlier mitigation in low-precision backpropagation. Since the first publication of this work, a number of alternative methods for FP4 pre-training have been proposed, either as concurrent alternatives [13], or extensions to Quartet [34; 10].

## 3 Background

**Quantization grids.** Quantization maps high-precision internal model states, such as weights, activations, or gradients, to a lower-precision discrete set—i.e., the *quantization grid*. This grid can be *uniform*, e.g., for integer quantization, or *non-uniform*, e.g., floating-point (FP) quantization, where the value spacing is roughly exponential for fixed exponent. Since the original values may differ in scale compared to the grid, a higher-precision *scale s* is typically stored alongside the quantized values. For a vector $x$, the quantization process can be written as $q(x) = \text{round}\left(\frac{x}{s}; \text{grid}\right)$, and the original values can be approximately reconstructed as $\hat{x} = s \cdot q(x)$. Common choices for the scale are setting it to the maximum absolute value (absmax) in $x$ (to avoid clipping) or optimizing it to minimize the mean squared quantization error, e.g. [37].

**Quantization granularity.** Apart from grid choice, quantization methods also differ in the *granularity* of the scales. A single scale value can be shared across an entire tensor, e.g. [3], across each row or column [37], or over more fine-grained custom-defined blocks, such as 2D blocks [57; 31] or 1D blocks [36; 47]. Notably, the latest Blackwell GPU architecture [35] introduces hardware support for

MXFP4/6/8 and NVFP4 formats. MXFP [36] formats share an FP8 power-of-two scale over each 1D block of 32 elements, while NVFP4 [35] uses FP8 (E4M3) scales and 1D blocks of 16 elements.

**Rounding.** Quantization typically involves rounding, e.g., via *deterministic rounding* to the nearest grid point, results in the lowest mean squared error (MSE). In contrast, *stochastic rounding* introduces randomness, rounding up or down with probabilities based on the input's distance to nearby grid points. While it may introduce higher MSE, stochastic rounding helps control bias, which can be crucial for maintaining the convergence of iterative optimization algorithms [1].

**Outlier mitigation.** One key issue when quantizing neural networks is the existence of large *outlier* values in the network weights, activations, and gradients [18]. One standard way of mitigating such outliers [43; 9; 3; 2; 47] is via the Hadamard transform: given a vector $x \in \mathbb{R}^d$, $h(x)$ is defined as $h(x) = H_d x$, where $H_d \in \mathbb{R}^{d \times d}$ is the normalized Hadamard matrix with elements from $\{\pm 1\}$. Hadamard matrices have a recursive structure $H_d = \frac{1}{\sqrt{2}} H_2 \otimes H_{d/2}$, which enables efficient computation when $d$ is a power of two [22]. Optimized FWHT implementations for GPUs are available [17; 44]. When $d$ is not a power of two, the input vector $x$ is typically either zero-padded to the next power of two or transformed using a *Grouped Hadamard Transform*, where $x$ is split into equal-sized blocks (each with power-of-two length), and the Hadamard transform is applied independently to each block.

**Blackwell Architecture Support.** NVIDIA's 5th-gen. Tensor Cores in Blackwell [35] provide native 4-bit floating-point execution. The cores support different block-scaled formats such as MXFP4 [36] and NVFP4 [35], which roughly double the peak throughput over FP8/FP6, with a single B200 GPU peaking at 18 PFLOPS of dense FP4 compute [35]. Interestingly, our investigation shows that, as of now, MXFP4 is the only microscaling format with support for all required layouts for both forward and backward multiplications in low precision on Blackwell [45]. Therefore, we adopt MXFP4 for our implementation. This format stores each value using 1 sign bit + 1 mantissa bit + 2-bits for exponent. Every group of 32 elements shares a common 8-bit scaling factor, represented with 8 exponent bits, and no bits for mantissa. Blackwell's 5th-gen. Tensor Cores handle the required on-the-fly rescaling in hardware, without the need for software-based rescaling at CUDA level. Additional details are provided in Section 4.4.

**LLM pre-training.** We pre-train Transformers [51] of the Llama-2 [46] architecture in the range of 30, 50, 100, 200 million non-embedding parameters across a wide range of data-to-parameter ratios raging from 25x (around compute-optimal [27]) to 800x (extreme data saturation). We additionally selectively scale the model size up to around 7 billion parameters to verify training stability. We train all models on the train split of the C4 [19] dataset and report C4 validation loss as the main metric. We use the AdamW optimizer [32] with weight decay of 0.1, gradient clipping of 1.0, a 10% LR warmup and cosine schedule. We identify the optimal LR for one of the small unquantized baseline models, scale it inverse-proportionally to the number of non-embedding parameters and reuse for every quantization scheme we evaluate. We present all hyper-parameters in Appendix A.2.

# 4 Quartet: Four Ingredients for "Optimal" Quantized Training

## 4.1 Ingredient 1: Comparing Quantized Training Approaches via their Induced Scaling Laws

The ability of LLMs to scale predictably with both model size and data across orders of magnitude is a cornerstone of the current AI scaling landscape [28]. Mathematically, this says that the expected loss is a function of model and data parameters, often described in the form of a parametric function. This function can be fitted on a set of training runs, and then used to determine the optimal computational training regime [27] or to extrapolate model performance [21].

In this paper, we investigate scaling laws relating evaluation loss to the precision in which the forward and backward passes are performed, denoted by $P_{forward}$ and $P_{backward}$, respectively. For this, we propose a scaling law of the following functional form:

$$L(N, D, P_{forward}, P_{backward}) = \left( \frac{A}{(N \cdot \text{eff}_N(P_{forward}))^\alpha} + \frac{B}{(D \cdot \text{eff}_D(P_{backward}))^\beta} \right)^\gamma + E, \quad (1)$$

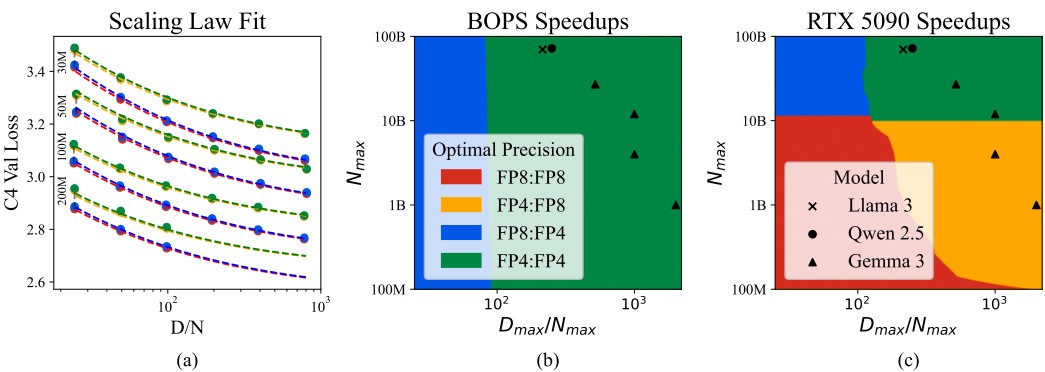

Figure 1: Analysis of Quartet: **(a)** Scaling-law 1 fit for various FORWARD:BACKWARD precisions. **(b)** Regions where each FORWARD:BACKWARD precision is optimal under the BOPS speedup model. **(c)** Same as (b) but with RTX 5090 speedups. Interestingly, popular models such as larger Llama3 or Qwen2.5 models fall into the FP4:FP4 optimality region, implying that training similar models in FP4 might have been optimal.

where $A, B, \alpha, \beta, \gamma$ are constants describing the general loss scaling w.r.t. model parameter count $N$ and training corpus size $D$.

The key addition is given by the fitted parameters $\text{eff}_N(P_{forward})$, representing the *parameter efficiency* of the precision $P_{forward}$ used in the forward pass, and $\text{eff}_D(P_{backward})$ representing the "data efficiency" of the backward pass occurring in a potentially different precision $P_{backward}$. (Both these factors are naturally in the interval $(0, 1]$, where the value 1 is reached for full-precision.) Specifically, our parametrization postulates that the impact of the forward-pass precision is felt primarily w.r.t. the trainable parameters, i.e., lowering precision to $P_{forward}$ lowers the model's "effective" parameter count to $N \cdot \text{eff}_N(P_{forward}) \leq N$. This follows the general trend of modeling the effect of forward pass quantization as a multiplicative factor on parameter count [25; 30; 26; 37]. For the data term, we postulate that lowering backward-pass precision primarily impacts the data term $D$, so we effectively need additional data to reach the same the same loss, precisely by a factor of $1/\text{eff}_D(P_{backward})$. This is a novel way to model backward pass quantization that we propose, consistent with optimization theory results [1], as well as observed performance gaps (see Figure 1 (a)). We present experimental data to justify these assumptions and compare against alternative scaling laws [30] in Appendix A.3.

Experimentally, we observe that different quantized training methods, e.g., STE [6] vs. QuEST [37], induce different scaling laws, and in particular different efficiency parameters. While, usually, scaling laws are used to extrapolate *model performance* across different parameter and data sizes, *we propose to use scaling laws to compare different training methods*. Specifically, we say that quantized training method A is superior to method B if it offers both higher parameter efficiency $\text{eff}_N$ and higher data efficiency $\text{eff}_D$.

## 4.2 Ingredient 2: Mixed-Precision Induces Inference-Training Trade-Offs

The above scaling law suggests that, given a set of scaling parameters and a target loss we wish the model to achieve, we can directly solve for the "optimal" forward and backward precisions which allow us to match the loss. However, as pointed out by Sardana et al. [41], it is often the case in practice that we wish to put a larger weight on inference cost, rather than training cost, which can lead to different results when determining the "optimal" training precisions. Because inference latency depends solely on the *forward* pass ($\sim 33\%$ of training compute) while the *backward* pass consumes the remaining $\sim 66\%$, these trade-offs may need to be analyzed separately.

Specifically, we can state a set of simple guiding principles:

- **Forward pass.** Low-precision induces a trade-off between reduced *parameter efficiency*, and increased inference speed: for instance, we could train a larger model in terms of parameters $N$, but quantize its forward pass to lower precision, and obtain a better trade-off. As such, $P_{forward}$ should be picked to optimize this trade-off.

- **Backward pass.** Similarly, *training speedup due to a quantized backward pass* can offset the *reduced data efficiency* eff$_D$: we could train more heavily-quantized model *on more data* under the same computing budget. Thus, $P_{backward}$ should be picked to optimize this trade-off.

We contrast this with previous work, which often requires lower precision to suffer *no* accuracy loss (e.g., Chmiel et al. [12]). This unnecessarily reduces these trade-offs to simple selection of the fastest lossless precision. We argue that scaling-law analysis enables a more fine-grained approach needed to decide upon the "optimal" set of forward and backward precisions.

**Example speedup model.** To illustrate this, we assume a hardware-agnostic bit-wise ops (BOPS) model, which states that speedup is inversely proportional to datatype bit-width. The speedups are stated in Table 1, relative to an FP8 baseline:

Then, given a forward-pass compute budget $N_{\max}$ and a training budget $N_{\max}D_{\max}$, the effective loss will be given by:

$$Loss\left(N_{\max}\ \mathrm{spfw},\ D_{\max}\ \mathrm{sptr}\,/\,\mathrm{spfw},\ P_{\mathrm{fwd}},\ P_{\mathrm{bwd}}\right),$$

which we evaluate with the scaling law from Equation (1), leading to the fit from Figure1(a). One can see how spfw and sptr propagate as multiplicative factors on eff$_N$ and eff$_D$ and directly counter the suboptimal parameter and data efficiencies. Figures 1(b)–(c) illustrate the optimality regions: specifically, it tells us for which model sizes (Y axis) and corresponding relative training compute (X axis) FP4 is optimal relative to FP8 (red vs. orange region). The green area is the region in which *training using our MXFP4 implementation* would be optimal by this metric. In Figure 5 we demonstrate that validation loss, on which we build the comparison, is consistent with downstream performance, meaning that the optimality propagates there as well.

Table 1: Speedups relative to an FP8 baseline for forward (spfw), backward (spbw); sptr is the harmonic mean of spfw and spbw with weights $1/3$ (forward) and $2/3$ (backward).

| Operation | FP4:FP8 | FP8:FP4 | FP4:FP4 |
|---|---|---|---|
| Forward / Inference (spfw) | 2.0 | 1.0 | 2.0 |
| Backward (spbw) | 1.0 | 2.0 | 2.0 |
| Training (sptr) | 1.2 | 1.5 | 2.0 |

In summary, Ingredient 2 says that *low-precision impact should be analysed under the compute budget*; scaling-law fits then reveal when a given precision is the optimal choice for either pass.

### 4.3 Ingredient 3: Minimal Forward-Pass Error and Error-Bias Trade-off

The above ingredients should allow us to determine the "best" quantized training method among existing approaches, focusing on the hardware-supported MXFP4 [36] format.

**Forward pass quantization.** As detailed in Section 2, existing QAT (forward-only) approaches can be split into "noise injection" [5] and "error-minimization" strategies, e.g. [37]. Focusing on the forward pass, by the above discussion (Ingredients 1 and 2), we seek the approach which maximizes the parameter efficiency factor eff$_N$. For this, we implement four standard schemes for QAT: 1) stochastic rounding (SR) with standard AbsMax per-group normalization [47]; 2) vanilla round-to-nearest (RTN) quantization with AbsMax per-group normalization; 3) learnable scale clipping (LSQ) with RTN quantization [20; 56]; 4) Hadamard normalization followed by RMSE-based clipping (QuEST) [37]. For fairness, we apply the Hadamard transform to weights and activations for each one of these schemes before quantization. We compare these approaches following Section 4.1: we train models using each technique, apply scaling law fitting, and register their resulting eff$_N$ factors. For additional information, we also show representations' mean-squared error (MSE) for fitting random Gaussian data. The results are provided in the first rows/columns of Table 2.

The results in Table 2 show that QuEST has the best parameter efficiency eff$_N$ among all existing methods. Moreover, eff$_N$ appears to correlate heavily with MSE, as suggested by Panferov et al. [37, 38]. Additionally, the results align with the analysis of Chmiel et al. [12] that determined deterministic RTN to always be preferable to stochastic rounding for the forward pass.

**Backward pass: a novel error-bias trade-off.** The above findings do not transfer to backward pass quantization, as optimization theory shows that unbiased gradient estimation is critical for

Table 2: Illustration of error-bias trade-off between different quantized forward and backward pass approaches. For the forward (given by the $\text{eff}_N$ metric) the best performing method is QuEST, correlating with superior MSE over Gaussian input data. By contrast, for the backward pass (the data efficiency $\text{eff}_D$), the best performing method is RTN, that balances quadratic error with magnitude alignment. This justifies our choice of method, which combines block-wise QuEST on the forward, with RTN on the backward pass.

| Rounding | $\text{eff}_N$ | MSE | $\text{eff}_D^*$ | Misalignment ( $1 - \mathbb{E}\left[1/S\right]$) |
|---|---|---|---|---|
| Stochastic Rounding AbsMax | 0.42 | $2.77 \times 10^{-2}$ | 0.88 | 0 |
| Round-to-nearest AbsMax | 0.59 | $1.37 \times 10^{-2}$ | **0.93** | $9.3 \times 10^{-3}$ |
| QuEST (Hadamard + RMSE) | **0.64** | $1.32 \times 10^{-2}$ | 0.83 | $1.3 \times 10^{-2}$ |

convergence, e.g. [1]. This leads to a trade-off between the error minimization we can obtain on the forward pass, and the bias induced over the backward pass for a given method. We study this trade-off via a novel analysis of gradient alignment between different quantization methods.

To study gradient bias, we follow the analysis of [48; 49], who studied RTN quantization with randomized rotations, approximated by the randomized Hadamard transform, which we denote by $\widehat{H}$. They show that, while RHT makes quantization unbiased *in direction*, it adds a bias *in magnitude*. To address this, they proposed an approach that makes RTN projections of post-RHT vectors unbiased, denoted by $Q$, via the following input ($X$) and randomness ($\xi$) specific group-wise rescaling factor $S$:

$$\mathbb{E}_\xi[Q(X,\xi)] = X \text{ if } Q(X,\xi) = S \cdot \text{RTN}(\widehat{H}(X,\xi)), \text{ where } S := \frac{\langle X, X \rangle}{\langle \widehat{H}(X,\xi), \text{RTN}(\widehat{H}(X,\xi)) \rangle}.$$

Unfortunately, their re-scaling is incompatible with coarse group-wise scaling of the MXFP4 format, so we cannot use it in practice. However, we can still use their approach to gauge the degree of misalignment for different quantizers by simply studying their corresponding expected value of $1 - \mathbb{E}\left[1/S\right]$, which we call the *projection magnitude misalignment*. This factor is presented in Table 2, along with the MSE across different schemes. Focusing on stochastic rounding (SR) vs round-to-nearest (RTN) with AbsMax, one can see that SR trades higher error for perfect alignment.

To connect those quantities with training dynamics, we analyze the cumulative effect of misalignment and error on backward quantization for a 30M-parameters Llama model. In Figure 2 **(a)** and **(c)**, we plot the alignment metrics–Cosine Similarity and Projection Magnitude Misalignment—for inter-layer activation gradients as a function of back-propagation "depth". We can again observe the trade-off between similarity and magnitude misalignment. Finally, Figure 2 **(c)** connects those quantities to final model quality (loss gap vs. full-precision model) for increasing data-vs-parameters.

We observe that RTN backward quantization consistently outperforms the alternatives, although the gap with stochastic rounding (SR) decreases as the training duration grows [2].

**Summary.** Our analysis outlines a new trade-off between parameter efficiency on the forward (equated with quantization MSE), and data-efficiency on the backward (which characterize with with a trade-off between error and misalignment). In the following, we will adopt a "best of both worlds" approach, aiming to perform a forward pass that minimizes MSE (based on QuEST [37]) together with an RTN backward pass that is balances quadratic error with and magnitude alignment. The novel challenge, which we address next, will be an extremely efficient GPU-aware implementation of such an approach.

### 4.4 Ingredient 4: Fast GPU Support for Accurate Quantized Training

**Quartet Overview.** We integrate our prior discussion into Algorithm 1 (also shown in Figure 3), which aims to perform accurate training while executing *all three* matrix multiplications of a linear

---

[2]In the first draft of the paper, an implementation bug caused large performance degradation for RTN backward quantization for longer training runs. This degradation led to our arriving at a different conclusion (SR superiority), which we now reject.

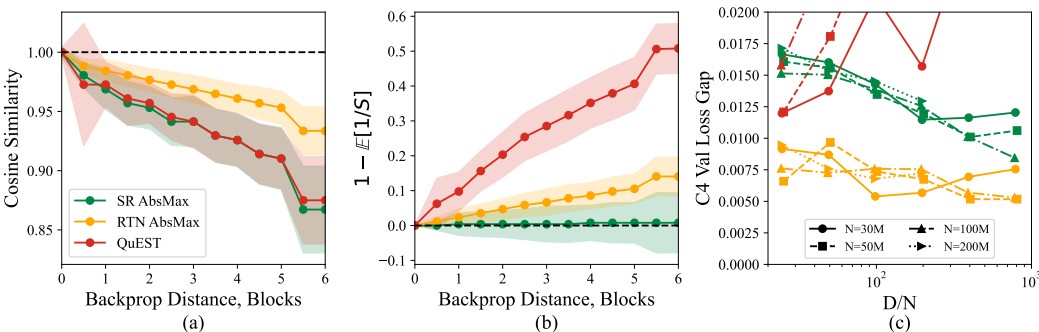

Figure 2: The effect of backward pass quantization on LLM training gradient quality and impact on performance: **(a, left)** and **(b, middle)** shows cosine similarity and projection magnitude misalignment with unquantized reference, while **(c, right)** shows performance gaps with a non-quantized baseline for a set model sizes and data-to-parameter ratios (D/N).

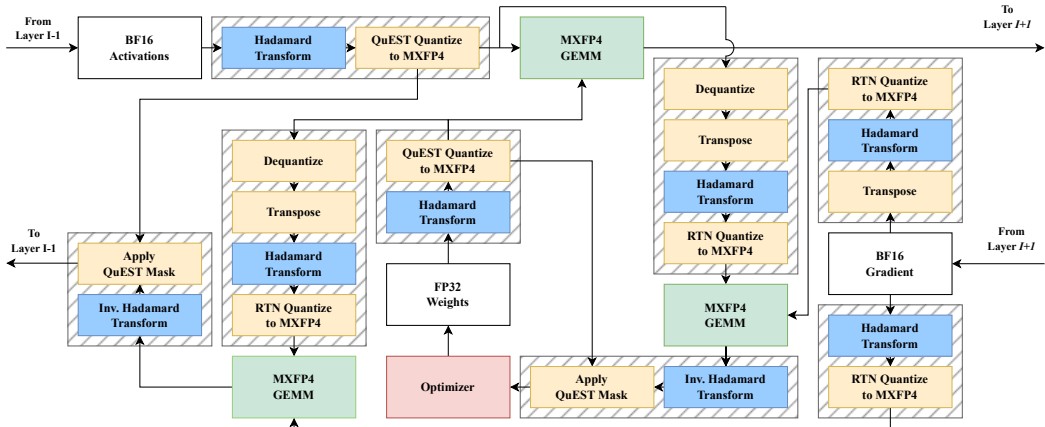

Figure 3: Quartet computation flow for a fully-MXFP4 linear layer training.

layer in low precision. The **forward pass** applies a fixed Hadamard transform $H_g$ (of block size $g$ equal to the quantization group size) and QuEST projection to low precision and multiplies the resulting tensors with an MXFP4 kernel. The **backward pass** decorrelates the multiplied tensors with an identical block-wise random Hadamard transform $\widehat{H}_g$, applies round-to-nearest (RTN) to MXFP4, performs the two gradient GEMMs in MXFP4, rescales to compensate for RTN range matching [47], applies QuEST masks ($M_x$, $M_w$) and inverts the Hadamard transform $H_g$.

**Costs and format specialization.** The key added cost of the above pipeline is that of the Hadamard rotations and their inversion: specifically, two Hadamard/Inverse transforms are added over standard training. Our key observation is that, since the MXFP4 already groups 32 consecutive weights (in 1D), sharing scales, we can and should apply the Hadamard rotations and their inversion at the same group size. With a fast Hadamard implementation, the theoretical cost is $O(g \log g)$—negligible for $g \leq 256$ compared with the GEMMs.

**GPU kernel support.** While the above blueprint appears simple, implementing it efficiently on Blackwell GPUs—in order to leverage fast MXFP4 support—is extremely challenging. For illustration, a direct implementation of the above pattern would be *slower* than FP16 unquantized training, let alone optimized FP8. Our fast implementation builds on CUTLASS 3.9 [45], which provides templates for the new Blackwell architecture. Computation happens in two stages: **Stage 1** fuses the Hadamard transform, quantization, scale calculation, and QuEST clipping mask generation (only on forward) into a single kernel; **Stage 2** performs GEMM using a dedicated kernel.

**Stage 1: Fused quantization-related operations**. First, we observe that, thanks to the small group size, the Hadamard transform can be implemented as a direct GEMM between the corresponding

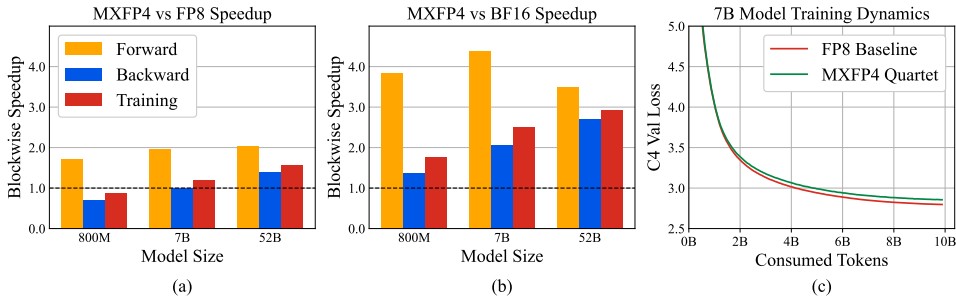

Figure 4: **(a, left), (b, middle)**: Quartet kernels block-wise speedup across model sizes relative to FP8 and BF16. **(c, right):** Training dynamics for the 7B model trained with Quartet relative to FP8 .

input matrix and a fixed $32 \times 32$ Hadamard matrix (see Sec. 3), producing output in FP32, which is stored in GPU Shared Memory (SMEM). This allows us to implement the Hadamard operation efficiently by leveraging CUTLASS's multilevel tiling templates to optimize data movement. All subsequent operations are integrated via a custom CUTLASS *epilogue*, which utilizes the intermediate results previously stored in higher levels of the memory hierarchy and operates locally in the Register File (RF). At this stage, Blackwell's new hardware support is used to downcast FP32 values to FP4 (E2M1) using the PTX instructions for this purpose. To construct the final MXFP4 format, we compute scaling factors of shape $1 \times 32$. These scales are represented in 8-bit using the E8M0 format. Finally, the clipping mask is computed, and the three resulting tensors (values, scales, and mask) are written to Global Memory (GMEM). Throughout, data storage is optimized to use the widest memory instructions possible.

**Stage 2: Dedicated GEMM kernel**. Blackwell introduces the `tcgen05.mma` instructions, which natively support matrix multiplication with scale factors in the form $D = C + (A \times \text{SFA}) \cdot (B \times \text{SFB})$. These scale factors are applied along the inner $(K)$ dimension of the GEMM. For MXFP types, every 32 elements along the $K$-dimension of matrices $A$ and $B$ share a corresponding scale factor. This implies that an $M \times K$ matrix $A$ is associated with a scale matrix SFA of size $M \times \lceil K/32 \rceil$. Our dedicated kernel is based on CUTLASS block-scaled GEMM for narrow precision. As part of this implementation, we also included the necessary functions to reorganize the scale factors generated in the Stage 1, aligning them with the layout required by this architecture [35].

To our knowledge, our implementation is the first to efficiently support quantization-related operations for microscaling formats on the Blackwell architecture. We release it as part of "QuTLASS", an open-source library that can be accessed here.

## 5 Experiments

We now provide additional experimental support for the validity of Quartet, shown in Figure 3, focusing on accuracy comparisons with existing INT4/FP4 training methods, and examining kernel speedups.

**Experimental setup and scaling law fit.** As described in Section 3, we pre-train Llama-style models on C4 and report validation loss after a fixed token budget. All baselines reuse the optimizer, schedule, and hyper-parameters, as described in Appendix A.2. Following Section 4.1, we compare accuracy across methods by fitting the full scaling law in Eqn. 1 across methods, as follows: we fit parameters $A, \alpha, B, \beta, E$ and $\gamma$ on a grid of baseline precision runs (FP8 forward, FP8 backward) shown on Figure 1(a). Then we fit the parameter and data efficiencies $\text{eff}_N$ and $\text{eff}_D$ separately for every forward and backward quantization scheme we evaluate. The law is fitted identically to prior work in this area [27; 30; 8]. For a more detailed description we refer to Appendix A.3.

**Accuracy comparisons.** We compare accuracy (validation loss) as well as the efficiency factors against four recent, fully–quantized training pipelines that operate in 4-bit precision for *both* forward and backward passes: 1) **LUQ** [12] applies to both INT4 and FP4, using unbiased quantization that pairs 4-bit weights/activations with stochastic underflow, and logarithmic stochastic rounding; 2) **HALO** [3], which uses Hadamard rotations to mitigate outliers, evaluated in FP4 at their most accurate HALO-2 setting; 3) **Jetfire** [58] performs quantization in blocks of $32 \times 32$, originally

introduced for INT8, and adapted to FP4 for our setup; 4) **LSS** [56] for INT4 training, that combines a Hadamard-based forward pass with "leverage–score" sampled INT4 gradients.

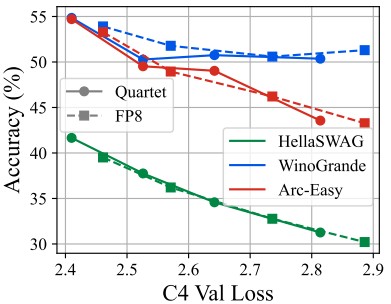

Figure 5: Correspondence between validation loss on C4 and various few-shot benchmarks for Llama models with 30-200M parameters.

Table 3: Validation loss (lower is better) on C4 for Llama models with 30M parameters and efficiency coefficients fitted on them. Columns show the tokens-to-parameters ratio ($D/N$). All methods share identical setups; only the quantization scheme varies. NaNs for LSS-INT4 appeared at arbitrary stages of training without any irregularities.

| Method | $25\times$ | $50\times$ | $100\times$ | $200\times$ | $400\times$ | $\mathrm{eff}_N$ | $\mathrm{eff}_D$ |
|---|---|---|---|---|---|---|---|
| LUQ–INT4 | 3.73 | 3.68 | 3.66 | 3.43 | 3.40 | 0.49 | 0.15 |
| LUQ–FP4 | 4.81 | 4.91 | 4.88 | 4.84 | 4.80 | 0.01 | 0.07 |
| Jetfire–FP4 | 7.03 | 6.94 | 6.76 | 6.62 | 6.58 | Unstable | |
| HALO–FP4 | 6.65 | 7.04 | 6.55 | 6.50 | 5.38 | Unstable | |
| LSS–INT4 | NaN | 3.40 | NaN | NaN | NaN | Unstable | |
| **Quartet** | **3.49** | **3.38** | **3.29** | **3.24** | **3.20** | **0.65** | **0.95** |

**Accuracy discussion.** As can be seen in Table 3, across all token-to-parameter ratios, Quartet attains the lowest loss, often by very large margins. At a tokens per parameter ratio of $100\times$, Quartet improves upon LUQ–INT4 by 10% relative loss, and the gap widens as we increase data size. We note that Jetfire and HALO incur large degradation and are unstable when ported to FP4. Interestingly, LSS is competitive only for shorter runs, and diverges for longer training budgets, beyond $50\times$, matching observations from prior work [23]. Overall, LUQ–INT4 is the strongest prior work; however, Quartet reaches significantly higher parameter and data efficiency, suggesting that it requires, roughly, 15% fewer parameters and 5x less data to reach the same loss. Figure 4 (c) additionally demonstrates the stability of Quartet for training models two orders of magnitude larger (7B parameters).

Additionally, we trained 100M, 200M, 430M, 800M and 1.6B parameters Llama models with Quartet and FP8, with $D/N = 100$. We evaluated them on a set of few-shot benchmarks, including HellaSwag [60], WinoGrande [40] and ARC-easy [15]. Figure 5 demonstrate that those evaluations are consistent with C4 validation loss for larger models.

**Speedup results.** Next, we evaluate the efficiency of our implementation on the NVIDIA RTX 5090 GPU by measuring its performance across single layers of standard shapes, and aggregating across an entire transformer block. Speedup results are shown in Figure 4, using a batch size $64$ and sequence length of $512$. The FP8 baseline is provided by CUTLASS MXFP8 kernels, while the BF16 baseline uses PyTorch, both using Blackwell-optimized kernels. Inference speedups are more pronounced due to the lower cost of the forward pass compared to the backward pass, and the latter's higher computational complexity. The speedup scales with the arithmetic intensity (i.e., model size), reaching up to $2\times$ over FP8 and $4\times$ over BF16 on the forward pass, where it stabilizes. In the backward pass, our implementation achieves up to $1.5\times$ over FP8 and $2.6\times$ over BF16, resulting in an overall training speedup of up to around $1.6\times$, and $2.9\times$, respectively.

## 6 Discussion and Limitations

We provided a set of guidelines to modeling, comparing and designing fully-quantized training schemes for large language models. Moreover, we followed those guidelines to arrive at Quartet: a new SOTA full MXFP4 training algorithm. One current limiting factor is that Quartet was designed with a specific (standard) data-type and compute architecture in mind. In future work, we plan to look into generalizing our approach to alternative formats, as well as larger-scale distributed model execution.

## Acknowledgments

This research was funded in part by the Austrian Science Fund (FWF) 10.55776/COE12, i.e., the Bilateral AI Cluster of Excellence, and through generous gifts by NVIDIA and Google. This work was supported under project ID 40 as part of the Swiss AI Initiative, through a grant from the ETH Domain and computational resources provided by the Swiss National Supercomputing Centre (CSCS) under the Alps infrastructure. We would like to thank our contacts at Datacrunch/Verda (Paul Chang and Antonio Dominguez) for hardware support that was essential to this project.

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

# A Technical Appendices and Supplementary Material

## A.1 Algorithm

---
**Algorithm 1** Quartet MXFP4 Forward-Backward Algorithm

---
**Require:** Hadamard Transform ($H_g$, $\widehat{H}_g$) block size $g$

1: **function** FORWARD(input $X$, weights $W$)
2:     $X_h \leftarrow H_g(X);\ W_h \leftarrow H_g(W)$
3:     $(X_q, M_x) \leftarrow \text{QuEST}(X_h)$
4:     $(W_q, M_w) \leftarrow \text{QuEST}(W_h)$
5:     $y \leftarrow \text{GEMM}_{\text{LP}}(X_q, W_q)$
6:     **return** $y$, ctx $= \{X_q, W_q, M_x, M_w\}$
7: **end function**

1: **function** BACKWARD(output gradient $dy$, ctx, seed $\xi$)
2:     Unpack $\{X_q, W_q, M_x, M_w\}$ from ctx
3:     $G_h \leftarrow \widehat{H}_g(dy, \xi);\ W_h^\top \leftarrow \widehat{H}_g(W_q^\top, \xi)$
4:     $G_q \leftarrow \text{RTN}(\tfrac{3}{4}G_h);\ W_q^\top \leftarrow \text{RTN}(\tfrac{3}{4}W_h^\top)$
5:     $dx_q \leftarrow \text{GEMM}_{\text{LP}}(G_q, W_q^\top)$
6:     $dx \leftarrow \tfrac{16}{9}H_g^{-1}(dx_q \odot M_x)$
7:     $G_h^\top \leftarrow \widehat{H}_g(dy^\top, \xi);\ X_h^\top \leftarrow \widehat{H}_g(X_q^\top, \xi)$
8:     $G_q^\top \leftarrow \text{RTN}(\tfrac{3}{4}G_h^\top);\ X_q^\top \leftarrow \text{RTN}(\tfrac{3}{4}X_h^\top)$
9:     $dW_q \leftarrow \text{GEMM}_{\text{LP}}(G_q^\top, X_q^\top)$
10:     $dW \leftarrow \tfrac{16}{9}H_g^{-1}(dW_q \odot M_w)$
11:     **return** $dx, dW$
12: **end function**

---

## A.2 Training Hyper-parameters

Table 4 lists model-specific hyper-parameters. Table 5 lists hyper-parameters shared across all experiments.

| Hyperparameter | 30M | 50M | 100M | 200M | 7B |
|---|---|---|---|---|---|
| Number of Layers ($N_{\text{layer}}$) | 6 | 7 | 8 | 10 | 32 |
| Embedding Dimension ($N_{\text{embd}}$) | 640 | 768 | 1024 | 1280 | 4096 |
| Attention Heads ($N_{\text{head}}$) | 5 | 6 | 8 | 10 | 32 |
| Learning Rate (LR) | 0.0012 | 0.0012 | 0.0006 | 0.0003 | $9.375 \cdot 10^{-6}$ |

Table 4: Model-specific hyperparameters used in our experiments.

| Hyperparameter | Value |
|---|---|
| Sequence Length | 512 |
| Batch Size | 512 |
| Optimizer | AdamW |
| Learning Rate Schedule | Cosine decay with 10% warm-up |
| Gradient Clipping | 1.0 |
| Weight Decay ($\gamma$) | 0.1 |
| Number of GPUs | 8 |
| Data Type (optimizer/accumulators) | FP32 |

Table 5: Common hyperparameters used across all model sizes and quantization setups.

## A.3 Scaling Law fitting

We fit the scaling law in two stages: **Stage 1.** Identical to prior work [8], we fit the unquantized scaling law of the form

$$L(N, D) = \left( \frac{A}{N^\alpha} + \frac{B}{D^\beta} \right)^\gamma + E$$

on baseline BF16 runs for $N \in [30M, 50M, 100M, 200M]$ and $D/N \in [25, 50, 100, 200, 400, 800]$ (see Figure 1 (a)) using Huber loss with $\delta = 10^{-4}$ on logarithm of $L$. Table 6 shows the resulting fit.

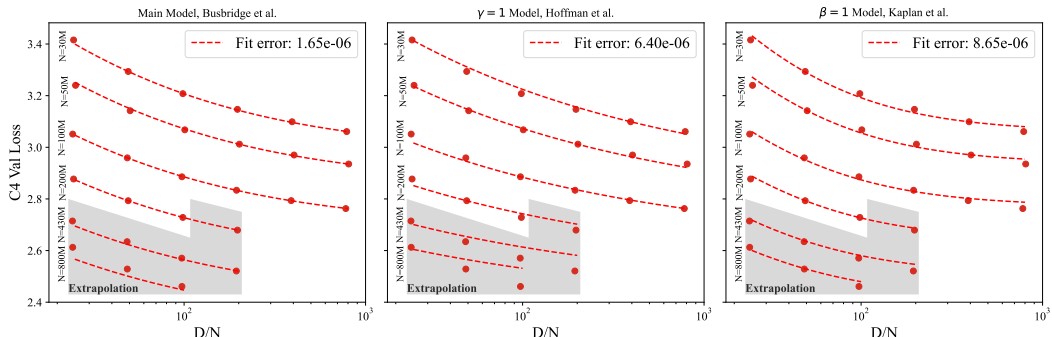

Figure 6: Comparison of various scaling law fits and their errors.

**Stage 2.** Using the fixed fitted parameters from **stage 1**, we fit the additional $\text{eff}_N$ and $\text{eff}_D$ parameters using the same loss function.

For the isolated methods compared in Section 4.2, we fit $\text{eff}_N$ and $\text{eff}_D$ independently for forward-only and backward-only quantization respectively.

For the end-to-end 4-bit comparison in Section 5, we fitted the parameters jointly for the setups present in Table 3.

**Alternative forms.** We additionally for the scaling law forms with fixed $\gamma = 1$ [27] and $\beta = 1$ [28]. The fits are presented in Figure 6 alongside the mainly used of Busbridge et al. [8].

| Parameter | $A$ | $\alpha$ | $B$ | $\beta$ | $E$ | $\gamma$ |
|---|---|---|---|---|---|---|
| Value | $1.52 \cdot 10^5$ | 0.589 | $5.25 \cdot 10^5$ | 0.544 | 1.35 | 0.274 |

Table 6: Fitted scaling law coefficients.

## A.4 Performance breakdown

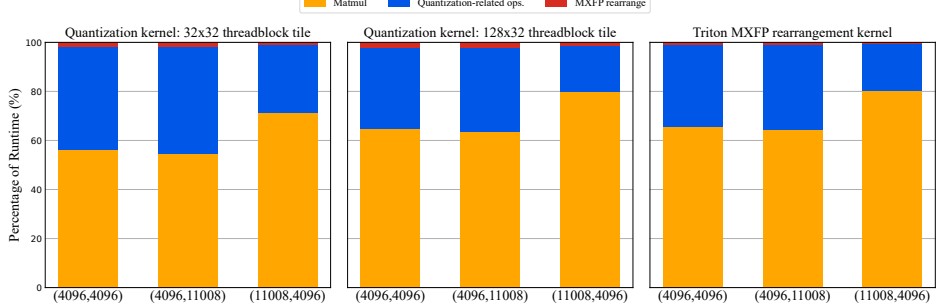

Figure 7: Breakdown of runtime composition across three linear layer shapes of a Llama-7B model, for an input of batch size $64$, and sequence length $512$.

Figure 7 presents a breakdown of runtime composition across three linear layer shapes in a Llama-7B model, taking the MXFP4 forward pass as an example. Each subplot shows the percentage of total runtime spent in three key kernel stages: matrix multiplication, quantization-related operations, and rearrangement of scaling factors for the `mma` instruction [35].

The figure compares three kernel configurations. The left subplot shows our fused kernel for quantization-related operations using a basic $32 \times 32$ threadblock tile size. The center subplot increases this tile size to $128 \times 32$, resulting in a more efficient quantization stage. The right subplot includes a custom Triton kernel, which further improves performance by optimizing the MXFP rearrangement stage. All results are normalized to $100\%$.

As the figure illustrates, tuning the quantization kernel significantly reduces the proportion of time spent in the quantization stage—particularly for large matrix shapes. Increasing the threadblock tile size leads to more active warps per block, enhancing arithmetic intensity and enabling better latency hiding. In CUTLASS-based implementations, this change influences the multilevel tiling strategy (threadblock, warp, and instruction-level tiling), which is designed to optimize data movement through shared memory and registers [45]. The Triton backend exhibits similar trends, with rearrangement overheads further reduced and matrix multiplication dominating the total runtime.

## A.5 End-to-end Prefill Speedups

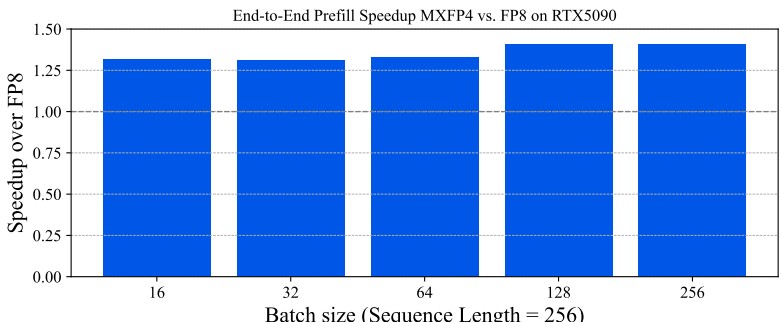

Figure 8: End-to-end prefill speedups for Quartet MXFP4 vs. FP8, across different batch sizes, using the 7B parameter model on a single RTX 5090.

Figure 8 illustrates the inference prefill speedup of MXFP4 over FP8 as a function of batch size, evaluated at a fixed sequence length of 256 on a 7B parameter model. The results demonstrate a consistent improvement in performance using MXFP4 across all batch sizes, with speedup increasing progressively and peaking at $1.41\times$ relative to FP8 at a batch size of 128, where it plateaus.

## A.6 Post-Training Quantization Results

We compare the results of applying post-training quantization (PTQ) against QUARTET using the MXFP4 format on the largest 7B model. For the PTQ baseline, we evaluate against QUAROT [2], where the weights are quantized using GPTQ [24]. To ensure a fair comparison, we introduce two key modifications to the original QUAROT approach:

1. **Attention Module:** We remove the use of online Hadamard transformations and instead apply a fixed Hadamard transformation of size 128 to the output dimension of the *v_proj* layer and the input dimension of the *out_proj* layer. This optimization accelerates the overall process by eliminating per-head online Hadamard computations, without affecting accuracy, since we use a group size of 32 in the MXFP4 format.

2. **MLP Down-Projection:** For *down_projection* layers with non-power-of-two dimensions in the MLP, we apply grouped Hadamard transformations using the largest power-of-two size that evenly divides the intermediate dimension of the MLP.

| Model Size | BF16 | QuaRot (PTQ) | Quartet |
|---|---|---|---|
| 7B | 16.40 | 18.19 | 17.77 |

Table 7: Perplexity results on C4 dataset using MXFP4 quantization. We use 128 samples from the training set (of the same dataset) as the calibration set in GPTQ.

Table 7 presents the comparison between the PTQ scheme (QuaRot) and QUARTET. QUARTET achieves a 0.42-point lower perplexity (PPL) compared to QuaRot when applied to the same model.

Notably, QUARTET is also more efficient than standard QAT methods, as it quantizes both forward and backward passes.

### A.7 Compute Resources

The pre-training experiments were conducted on datacenter-grade machines with 8xH100 NVIDIA GPUs for a total compute of around 6,000 GPU-hours. Although most experiments do not require such an elaborate setup, we found the 7B pre-training experiment specifically to be very DRAM-demanding and to require such specific hardware.

The speedup results were obtained on a consumer-grade NVIDIA RTX5090 GPU with total runtime of under 1 hour.

