# OpenReview forum: "Quartet: Native FP4 Training Can Be Optimal for Large Language Models"
_NeurIPS.cc/2025/Conference — NeurIPS 2025 poster_

### Official Review · Reviewer_7M4U · 2025-06-30

**Clarity:** 2
**Significance:** 3
**Originality:** 3
**Rating:** 5
**Confidence:** 3

**Summary:**

This paper introduces an algorithm called Quartet, which allows for FP4 training on the NVIDIA Blackwell architecture. Background and motivation is provided, including related works. In order to develop Quartet, the authors claim 4 contributions. A scaling method comparing quality parameters for quantized training methods, a way to relate these quality metrics to concrete training algorithms via certain [novel] methods, an optimized implementation of Quartet on modern hardware, and experimental evaluation against previous baselines.

**Questions:**

### Major

Can you clarify how you arrived at the scaling law in section 4.1?

Can you clarify section 4.2, especially from lines 233 to the effective loss formula?

Why was LSQ omitted in Table 2?

### Minor

The NVIDIA 5090 GPU is a slightly different compute capability to B200 and GB200 cards. Are there any differences here in terms of the FP4 capabilities?

As the paper is focused on using a specific accelerator architecture (NVIDIA Blackwell) why not focus on NVFP4?

In line 387, how are the baselines implemented? For example, are they both using stochastic rounding? Is this actually using a framework like Megatron or Deepspeed?

Were any loss spikes present during training?

Figure 1: Isn’t the scaling law fit dependent on the training method used for those precisions? Why isn’t this specified?

[1] NVIDIA, CUDA GPU Compute Capability, https://developer.nvidia.com/cuda-gpus

**Ethical Concerns:**

["NO or VERY MINOR ethics concerns only"]

**Final Justification:**

This paper is a solid addition to the area of low precision training, which is a timely and important topic. The paper is clear and well written, while being novel, significant and overall well justified by experiments and theory. I believe this will be of interest to many machine learning researchers and practitioners.

**Limitations:**

Yes.

**Paper Formatting Concerns:**

None.

**Quality:**

3

**Strengths And Weaknesses:**

I would first like to thank the authors for submitting their work to NeurIPS 2025.

### Pros

This paper targets a relevant and important area today, which is low precision/quantized training. The paper starts with a well written problem statement and primer to low precision training and inference. The claimed contributions are clearly stated.

Overall, the related work section also gives a concise but useful description of prior works.

Background sections on quantization grids, quantization granularity, rounding, architecture support, and pre-training are clear and informative.

Section 4 is again well written in a smart manner with the four different “ingredients” of Quartet. The structure of the scaling law in 4.1 at least makes intuitive sense as well.

Section 4.3 experiments are well constructed and targeted and overall easy to understand. This is primarily due to the introduction in this section of the misalignment metric, as well as Table 2 which makes it easy to see the trade off between $\text{eff}_N$ and $\text{eff}^*_D$ for the different methods.

Section 4.4 is clearly well thought out, and represents a significant contribution given the non-triviality of implementing these kinds of low precision operators in an efficient way on hardware. Also, Algorithm 1 is compact but descriptive, and I feel that it would help if someone were to independently attempt to reproduce the implementation.

Although the evaluations are quite concise, it is satisfactory in terms of validating the method presented, going over speedup, loss curve, $\text{eff}_N$/$\text{eff}_D$ improvement over previous methods.

### Cons

Lines 93-94: Treating FP8 as the lossless baseline is fine, but the reason should be clearly stated in that sentence. There is also no discussion of “master weights” techniques, which appears to have been used for MXFP4 previously [1] that some readers may be familiar with.

In section 3, outlier mitigation, it isn’t clear why the grouped hadamard transform helps. Although there is some background on the transform itself, it is less clear how this translates to helping with outliers.

In section 4.1, it should be made clear that the general form of scaling-law (1) is built on prior work [8] in the main paper as opposed to the supplementary material and that that form is being assumed for that reason.

Section 4.2 was quite confusing. My understanding is, In lines 233-234, $N_{max}$ is supposed to be the forward pass parameter budget, which influences the compute budget, but isn’t the compute budget itself. And the training budget similarly is based on $D_{max}$. Speedups are clear, but it would be better to show how $\text{spfw}$, $\text{spbw}$, $\text{sptr}$, are functions of $P_{fwd}$ and $P_{bwd}$ given this simplified BOPS setting. Some derivations here would be good.

Section 4.3 should include some information on the experimental setting itself. Also, it’s not clear why LSQ is omitted from Table 2.

Although it’s mentioned as a future extension, it would be good to include at least some discussion regarding larger model sizes. 7B is on the smaller size today, and it’s not clear how Quartet (especially the scaling law) would generalize to larger model sizes where the same scaling may not hold.

The evaluation doesn’t include any other model architectures (mainly MoE), but this can be excused due to (presumably) the difficulty of writing those kernels.

---

Overall, I find that the work is polished, and its significance, originality, and quality to be high. I have some issue with the clarity in the (fairly critical) sections 4.1 and 4.2 which the majority of the paper relies on.

[1] Rouhani et. al, Microscaling Data Formats for Deep Learning, https://arxiv.org/pdf/2310.10537

#### Other Comments

I would have liked to see some experiments on the memory footprint of the new, lower precision implementation. But it’s fine not to include since that’s not the focus of the paper.

Nits:

Line 148: To be pedantic: *normalized* Hadamard matrices have that recursive property. Good to repeat there.

---

> ### Author Rebuttal · Authors · 2025-07-30
>
> We thank the reviewer for providing an extensive evaluation of our work and provide the requested clarifications:
>
> > Can you clarify how you arrived at the scaling law in section 4.1?
>
> Conceptually, we proposed this form to unify the laws proposed for forward-only compression by Kumar et al. [4], and the theoretical analysis of training with noisy gradients of e.g. QSGD [2], which suggest a separation of the impact of low precision on the forward and backward passes, in the parameter and data terms, respectively.
> Practically, the ablation for the general form is presented in Appendix A.2 in the supplementary material. In short, the A,B,E,$\alpha$,$\beta$,$\gamma$ form had by far the lowest fit error and was originally tested (by Busbridge et al.[1]) for models up to 7B parameters. This verified that it’s accurate and robust. The inclusion of eff$_N$ models fundamental reduction in capability, modeled similarly in numerous works cited around line 193 of our submission. On the contrary, eff$_D$ implies that the quantized gradient estimation allows for (albeit slower) convergence to the same loss, which aligns with the theoretical analysis of noisy gradient training [2], and the closing of the performance gap on Figure 1 (a).
>
> > Can you clarify section 4.2, especially from lines 233 to the effective loss formula?
>
> Assuming baseline FP8 training and inference, parameter count $N_{max}$ corresponds 1 to 1 to a certain inference cost (latency). Additionally, under fixed model size, training compute can be expressed in training corpus size $D_{max}$ (compute ~ $N_{max} \cdot  D_{max}$) or data saturation ratio $D_{max}/N_{max}$. Combined, these two quantities constitute a bias in the model cost functional space. Assuming certain inference and training speedups (spfw, sptr) for a certain training scheme and model sizes, various schemes can be mapped into that space for iso-compute comparisons. The effective loss formula presents cost-normalized parameter count and corpus size. Figure 1 (b, c) presents the regions in that space where certain precisions yield lower loss.
>
> > Why was LSQ omitted in Table 2?
>
> LSQ was shown by Panferov et al.[3] to consistently yield higher loss and ~20% lower eff$_N$ than QuEST. We chose the more modern and better performing method as the candidate. Additionally, LSQ can’t be directly applied to backward pass quantization as it has learnable parameters of its own. We will clarify this.
>
> > The NVIDIA 5090 GPU is a slightly different compute capability to B200 and GB200 cards. Are there any differences here in terms of the FP4 capabilities?
>
> Yes. According to the Blackwell data sheet [4], we have that:
>  - CC=120 (RTX 5090) compared to CC=100 (B200) has a higher relative speedup of FP4 compared to FP8 (4x in theory vs 2x in theory).
>  - CC=120 (RTX 5090) compared to CC=100 (B200) doesn’t offer a stochastic rounding instruction.
>
> The first fact means that FP4 would have a potentially lower performance “ceiling” on the B200 relative to the 5090. These aspects were not well-documented at the time of submission. We’ll make sure to include them in the discussion. We are currently working on a B200 version of our kernels.
>
> > As the paper is focused on using a specific accelerator architecture (NVIDIA Blackwell) why not focus on NVFP4?
>
>
> We chose MXFP4 specifically to not focus exclusively on Blackwell: MXFP4, being the standard OCP Microscaling Format, will see wider adoption among AI accelerator hardware. For instance, it will be supported on AMD hardware as well, whereas NVFP4 will (most likely) not be supported.
>
>
> At the same time, our kernel design does apply to NVFP as well. Specifically, we have already extended our forward kernels to NVFP4 (see response to reviewer **dMMD**) and plan to do so for the backward pass as well.
>
> > In line 387, how are the baselines implemented? For example, are they both using stochastic rounding? Is this actually using a framework like Megatron or Deepspeed?
>
> The baselines are implemented as either custom GEMM CUDA kernels for FP8 or basic `torch.nn.functional.linear` for BF16.
> For hardware support reasons discussed above, RTN instructions were used instead of stochastic rounding.
> The speedups are reported for quantized forward and backward passes over linear layers that would constitute models of certain size, effectively omitting non-linear operations and optimization steps from the comparison. They are expected to have negligible impact for larger models.
>
> > Were any loss spikes present during training?
>
> We observed no spikes in either loss or gradient norm. Figure 3 (c) shows a sample of the training dynamics.
>
> > Figure 1: Isn’t the scaling law fit dependent on the training method used for those precisions? Why isn’t this specified?
>
> We acknowledge this question and will clarify. The parameter and data efficiencies eff$_N$ and eff$_D$ indeed depend not only on the precision but generally the quantization scheme (method+precision). That’s why it can be used to both compare different methods and different precisions for the same method. We’ll adjust the wording to make it clearer.
>
> [1] https://arxiv.org/abs/2502.08606 [2] https://arxiv.org/abs/1610.02132 [3] https://arxiv.org/abs/2502.05003 [4] https://images.nvidia.com/aem-dam/Solutions/geforce/blackwell/nvidia-rtx-blackwell-gpu-architecture.pdf

---

> > ### Comment · Reviewer_7M4U · 2025-08-05
> > **Acknowledgment and Response**
> >
> > Thanks to the authors for providing this rebuttal. I have considered this and other reviews and responses. I appreciate the clarifications provided. It would be helpful for the paper to add some of these clarifications to the final versions, specifically the motivations around section 4.
> >
> > I am tentatively raising my score pending the end of the rebuttal period.

---

### Official Review · Reviewer_dhqV · 2025-07-03

**Clarity:** 2
**Significance:** 3
**Originality:** 2
**Rating:** 4
**Confidence:** 2

**Summary:**

This work presents Quartet, an end-to-end FP4 training recipe for all linear layers. Achieves state-of-the-art accuracy for billion-scale LLMs using FP4 only, measured by the data and parameter efficiency in the compute optimal scaling law proposed by this work. The scaling law that models how accuracy scales with forward/backward pass precision, capturing both parameter efficiency and data efficiency. Specifically, it:
- in the forward pass: Uses QuEST quantization (based on Hadamard transform and RMSE clipping) for minimal quantization error.
- in the backward pass: Uses stochastic rounding to reduce gradient bias and improve convergence in long training runs.

With efficient GPU Implementation, it leads to 1.8× training speedup overall compared to fp8, with higher accuracy than prior FP4/INT4 methods (e.g., Jetfire, HALO, LUQ) on model size up to 7B.

**Questions:**

- what is the portion of time spent on attention forward/backward operations in your benchmark, and if they're done in bf16, or fp8, or fp4?

**Ethical Concerns:**

["NO or VERY MINOR ethics concerns only"]

**Final Justification:**

My concerns are partially addressed. However, the context window 512 is much shorter than typical foundation model pre-training which requires at least 4k or 8k. I think overall this work has good merit but the speed up factor may not be as high as claimed in the paper in real world scenarios.

**Limitations:**

- limited to the llama architecture. not verified on MoE, or other non-transformer architectures.

**Quality:**

3

**Strengths And Weaknesses:**

strength:
- novel framework for analyzing the performance of low-precision algorithm via scaling law.
- good convergence result with fp4
- it shares the insight that the backward pass may be impacted by bias in gradient estimation and it's important to correct it
- if the code will be open sourced as stated, it will be quite helpful for the community

weakness:
- lack of clarity: does quartet perform the all attention matmuls in mixfp4 as well?
- insufficient hyper-parameter and model architecture sensitivity analysis: it looks like the experiment only involve the llama architecture, which is quite limited. It's covered in the literature that model architecture affects the quantization performance, for instance swiglu activation, or the position of the norm layers [1][2]. It's unclear if the proposed method works across different model architecture variant other than llama.
- it would be helpful to introduce the definition of mxfp4 format, for audience who is not familiar with the number representation
- So far the exploration for fp8 attention training is still ongoing. It's unclear if there's any caveat when applying mxfp4 to the attention layer, especially for long sequences training. The context length used for training is not specified in the paper either.

[1] https://arxiv.org/abs/2405.19279
[2] https://arxiv.org/abs/2409.12517

---

> ### Author Rebuttal · Authors · 2025-07-30
>
> We are grateful to the reviewer for their work.
> To address their concerns, we highlight that we performed quantization of linear operation in transformers, retaining attention operations in high (BF16) precision. More specifically, we isolated the quantization logic inside the linear layers, retaining all the remaining operations in the architecture in their original form.
>
> This was motivated since:
>  - Linear layers’ are the most computationally expensive part of LLM training and inference for short-to-medium context training and inference.
>  - Work on low-precision attention and efficient long-context modeling is orthogonal to linear layer speedups.
> The training sequence length of 512 tokens that we utilized and the subsequent omission of the attention operation from individual latency benchmarks reflects this choice.
>
> We acknowledge this limitation of our work, and we will include it in the discussion, but we emphasize that we consider the effect of attention quantization and its impact on long-context training an orthogonal research direction. (In fact, since submission time, orthogonal work on FP4 attention was posted independently [5].)
>
> *Regarding architectural verification*, we note that our setup closely follows experimental design of prior work on precision scaling laws, notably Kumar et al. [1], Panferov et al. [2], Tseng et al. [3] and Busbridge et al. [4].
> All these references focus on dense (non-MoE) Llama-like transformers.
>
> We emphasize that the proposed principles for scaling-law-based and iso-compute comparisons can be directly applied to other model architectures, such as MoE, but that would require designing, verifying and fitting of novel scaling law forms. As such, we leave this for future work.
>
>
> [1] https://arxiv.org/abs/2411.04330 [2] https://arxiv.org/abs/2502.05003 [3] https://arxiv.org/abs/2502.20586 [4] https://arxiv.org/abs/2502.08606
> [5] https://arxiv.org/abs/2505.11594

---

> > ### Author Response · Authors · 2025-08-07
> > **Call for discussion**
> >
> > Dear reviewer **dhqV**.
> >
> > As the discussion period is drawing to a close, we would like you to let us know whether your concerns were addressed and if there are any further clarifications we can provide. Thank you for your time and feedback.

---

> > > ### Comment · Reviewer_dhqV · 2025-08-07
> > >
> > > Thanks for the rebuttal. Thanks for the clarification of the sequence length used in this paper. However, the context window 512 is much shorter than typical foundation model pre-training which requires at least 4k or 8k. I think overall this work has good merit but the speed up factor may not be as high as claimed in the paper in real world scenarios. Therefore I'll keep my score.

---

> > > > ### Author Response · Authors · 2025-08-09
> > > > **Acknowledgment**
> > > >
> > > > Thank you for the reply! We commit to providing additional results for larger sequence lengths in the next revision of our work.

---

### Official Review · Reviewer_PgVh · 2025-07-03

**Clarity:** 3
**Significance:** 3
**Originality:** 3
**Rating:** 4
**Confidence:** 4

**Summary:**

The authors propose quartet, a recipe to train models. Furthermore, they propose parameter efficiency and data efficiency metrics, in tight relation to the forward pass precision and backward pass precision. They utilize stochastic rounding and MXFP4 format on the latest Blackwell GPUs and attempt to come up with a pure MXFP4 based training recipe.

**Questions:**

1. Do you have data points for Table 3 with FP4-FP8, FP8-FP4, FP8-FP8 precisions in the forward and backward passes respectively?

**Ethical Concerns:**

["NO or VERY MINOR ethics concerns only"]

**Final Justification:**

I have read the other reviews and the rebuttal by the authors. My score stays the same due to speedup concerns over fp8 baseline.

**Limitations:**

Yes.

**Quality:**

3

**Strengths And Weaknesses:**

Strenghts:
* The paper utilizes the strengths of existing methods, such as utilizing Quest in the forward pass and stochastic rounding in the backward pass.
* The parameter efficiency and data efficiency definitions make sense and the paper is sound overall.

Weaknesses:
* The comparison is focused on methods that solely utilize 4-bit precision, Table 3 could have included an MXFP8 baseline so that we can see how much quality loss is incurred by choosing to use 4-bit precision matmuls everywhere.

---

> ### Author Rebuttal · Authors · 2025-07-30
>
> We would like to clarify the use of FP8 as a baseline in this paper:
>
>  - From the accuracy perspective, we reported the BF16 results as an upper bound on FP8 accuracy whenever we needed to compare against FP8. This is a more-than-fair comparison, since FP8 methods do not outperform BF16 precision training, while some methods were shown to match it.
> We thus believe that this is a fair comparison against the best accuracy results FP8 training could possibly offer.
>
>
>  - From the speedup perspective, our intention was to compare against an ideal FP8 baseline method that only performed FP8 cast + GEMM, which would ignore any overheads due to error mitigation and would represent the fastest these FP8 kernels can be.
>
> However, after the submission deadline, we have realized that our evaluation induced a small but unnecessary performance overhead in the FP8 benchmarking relative to the ideal version described above.
> After repeating all the runs for the FP8 baseline, the impact of removing this overhead is the following:
>
> - On the smaller (800M) model, the ideal FP8 version reaches 20% higher inference throughput than in our submission, and 25% higher training throughput.
> For instance, our inference speedup relative to FP8 for this model size is reduced 1.9x to 1.6x.
> - On the larger (52B) model, the ideal FP8 version reaches **the same** inference throughput as in our submission, and <= 10% higher training throughput.
> Our end-to-end speedup stays roughly the same at this model size.
>
> We apologize for this error, but stress that the performance difference is small enough that it does not significantly affect our FP4 optimality claims: in particular, the optimal training precision for the Llama and Qwen models would still be FP4, whereas the Gemma3 model remains borderline.
>
>
> To sum up, our FP8 baseline aimed to **upper bound both the accuracy and the fastest possible performance of FP8 training**, representing the performance of an ideal FP8 training method.
>
>
> We thank the reviewer for their perspective, apologize for the error, and promise to add a more thorough explanation of our comparison setup and its implications.
>
> > Do you have data points for Table 3…
>
> All the data points and the code to produce all the figures, including Figure 1 (a) that shows the data you requested, are present in `./notebooks/plots.ipynb` in the supplementary materials.

---

> > ### Author Response · Authors · 2025-08-07
> > **Call for discussion**
> >
> > Dear reviewer **PgVh** .
> >
> > As the discussion period is drawing to a close, we would like you to let us know whether your concerns were addressed and if there are any further clarifications we can provide. Thank you for your time and feedback.

---

### Official Review · Reviewer_dMMD · 2025-07-07

**Clarity:** 3
**Significance:** 4
**Originality:** 3
**Rating:** 6
**Confidence:** 5

**Summary:**

The paper proposes Quartet and claims it is the first to enable a full FP4 training pipeline for large language models ensuring numerical stability and delivering performance faster than FP8 on NVIDIA’s new Blackwell GPUs. The authors present a precision-aware scaling-law framework that introduces two key concepts: The "parameter efficiency" concepts that is linked to the forward-pass error (eff_N) and the "data efficiency" concepts that is linked to the backward-pass gradient bias (eff_D).  Based on these concepts, Quartet was designed with these motivations and techniques:

- Forward-pass error reduction: The authors apply a block Hadamard rotation plus QuEST clipping before quantising, spreading outliers so weights/activations fit cleanly in 4 bits.
- Backward pass unbiased gradients: They use the same Hadamard decorrelation for backward tensors, but apply stochastic rounding to eliminate gradient bias and stabilize the learning.
- FP4 assessment: They derive a lightweight precision-aware scaling law that predicts final accuracy from model size, data volume and bit-width, revealing the regimes where FP4 could outperform FP8.
- Efficient kernel-level implkementation. They implement fused CUDA kernels (rotation to quantisation to scaled MXFP4 GEMM) that leverage Blackwell’s new FP4 tensor-core instruction, delivering ≈2× FP8 throughput. Using  custom CUTLASS 3.9 kernels, they fuse rotation, quantisation and MXFP4 GEMMs via the tcgen05.mma instruction.

The empirical evaluation demonstrates that Quartet was able to train Llama-style models up to 7 B parameters on C4 with no loss relative to FP8 while delivering about 2.4 × forward and 1.6 × backward kernel speed-ups.

**Questions:**

- Regarding the PMA proxy limits, can you provide intuition or data showing that PMA continues to correlate with convergence for larger models (> 7 B) or other tasks?
- How would Equation (1) break if quantization error interacts non-linearly with model width or depth?
- Your analysis of the error-bias trade-off (Figure 2c) suggests that stochastic rounding (SR) on the backward pass, as used in Quartet, is optimal for data-saturated regimes (high D/N), while round-to-nearest (RTN) may be better for compute-optimal regimes. Does this imply Quartet is primarily tuned for the "long training" setting? Have you considered an adaptive strategy, perhaps starting training with an RTN-based backward pass and switching to SR later, to potentially achieve optimality across all data budgets?
- The implementation of Quartet is specialized for the MXFP4 format. The Blackwell architecture also introduces the NVFP4 format. Could you comment on the challenges and potential performance implications of adapting Quartet to NVFP4?
- Table 3 shows that the LSS-INT4 baseline becomes unstable and diverges on longer training runs. Can you leverage your gradient analysis framework to provide any insights that might explain why leverage-score sampling on the gradients would lead to such instability?
- Great speedup results in (Figure 3). To better understand the overheads, can you provide a more granular breakdown of the execution time for a typical linear layer? Specifically, what is the latency contribution of the fused quantization kernel (Stage 1, including the Hadamard transform) versus the GEMM kernel itself (Stage 2)?
- The Quartet algorithm is a complex pipeline involving multiple components (Hadamard transforms, QuEST, stochastic rounding). This complexity might limit broad adoption or modification by researchers who are not experts in low-level GPU programming. Have you considered providing a higher-level Triton implementation (or at least Triton reference kernels) to lower the entry barrier?

**Ethical Concerns:**

["NO or VERY MINOR ethics concerns only"]

**Limitations:**

yes

**Quality:**

4

**Strengths And Weaknesses:**

Strengths:
- A technically well-written and structured paper. It was easy to read and follow.
- Clear demonstration of stable 7 B FP4 training curves.
- Hardware measurements demonstrated per-kernel speed-ups and back up the kernel-design claims.
- Scaling-law framework is fitted to data and not just asserted in the paper.
- The related work section gives a very good coverage of the INT8 → INT4 training paradigm (SwitchBack, JetFire, HALO, INT4-Transformers/LSS, LUQ). It clearly separates “noise-injection” vs. “error-minimization” QAT methods and shows how the forward path builds on them. It also connects algorithmic needs to hardware by highlighting Blackwell’s MXFP4/NVFP4 support. Citations are current (2023–2025) and well grouped, which makes the survey easy to follow.

Weaknesses:
- Only C4 perplexity is reported; no downstream or instruction-tuning tasks.
- The accuracy table limited to 30 M parameters; larger models lack final-epoch metrics and variance analysis.
- Speed-up numbers are kernel-level, there are no end-to-end wall-clock or energy-per-token measurements.
- No statistical error bars or standar deviation. The single-run results may not be statistically significant.
- Baselines sometimes diverge. It is unclear if with better tuning the authors might close the gap.
- The optimizer still uses FP32 so the “end-to-end FP4” efficiency claims are not fully validated.
- A big portion of the work performance advantage is derived from the highly specialized kernel’s implementation based on the NVIDIA Blackwell architecture and its MXFP4 format. While this makes the work practical/ impactful for this type of hardware, it also presents a limitation. The exact implementation and performance gains may not be directly transferable to other hardware platforms without similar native support for these specific operations.
- The proposed scaling law is an empirical model fitted on a specific architecture (Llama) and dataset (C4). The paper did not look at downstream tasks or other models. The idea that backward-pass precision primarily affects the data efficiency term is compelling but would benefit from further theoretical grounding or broader empirical validation across more diverse tasks and model families.
- In related work section, there are some notable omissions that would be good to add:
   - Native FP8 training (Micikevicius ’22, Fishman ’24). This critical because FP8 is the paper’s “lossless” baseline.
   - Optimizer-state compression (8-bit Adam, ZeRO-3 8-bit states) is relevant to any “end-to-end FP4” claim.
   - Foundational stochastic-rounding papers (Gupta ’15; DoReFa-Net ’16)  would complete the historical context for SR.
   - Sparsity/quantisation scaling laws (Frantar ’24, ’25)  are closely related to this paper’s precision scaling law.
   - Extreme low-bit LLMs (BitNet, 1-bit)  work is useful to situate Quartet against the lower-than-4-bit frontier.
   - SmoothQuant / AWQ equalisation  are other mainstream outlier-mitigation routes beyond Hadamard rotations.
   - FP8 kernel / Hopper Tensor Core literature. Adding a short comparison would clarify how the paper’s Blackwell FP4 kernels improve over existing FP8 paths.

---

> ### Author Rebuttal · Authors · 2025-07-30
>
> We thank the reviewer for constructive feedback and address their questions in their original order.
>
> > Only C4 perplexity is reported…
>
> We note that we determine optimality based on pair-wise evaluation loss comparisons. That means that as long as downstream (e.g. zero-shot) performance is consistent with loss, the comparisons remain valid. To demonstrate this consistency, we present 0-shot evaluations of the trained model in the table below:
>
> | N     | D    | Method  | C4 Val Loss | HellaSWAG | WinoGrande | Arc‑Easy |
> |-------|------|---------|-------------|-----------|------------|----------|
> | 1600M | 160B | Quartet | 2.410       | 41.66%    | 54.85%     | 54.71%   |
> | 800M  | 80B  | Quartet | 2.526       | 37.75%    | 50.27%     | 49.53%   |
> | 430M  | 43B  | Quartet | 2.642       | 34.58%    | 50.75%     | 49.03%   |
> | 200M  | 20B  | Quartet | 2.814       | 31.26%    | 50.36%     | 43.56%   |
> | 800M  | 80B  | FP8     | 2.461       | 39.51%    | 53.91%     | 53.28%   |
> | 430M  | 43B  | FP8     | 2.571       | 36.21%    | 51.78%     | 48.95%   |
> | 200M  | 20B  | FP8     | 2.736       | 32.77%    | 50.59%     | 46.21%   |
> | 100M  | 10B  | FP8     | 2.886       | 30.21%    | 51.30%     | 43.27%   |
>
>
> One can see that the zero-shot results show that the models provide non-random performance, and are consistent with the evaluation loss across the training methods.
>
>
> > Regarding the PMA proxy limits…
>
> The effect can be observed from two sides:
>  1. The statistical effect of magnitude alignment.
>  2. The fact that RTN is better for shorter training and SR - for longer.
>
> The first effect is expected to persist regardless of model size as it is a statistical property of quantized matrix multiplication during the backpropagation. The latter can be observed in the training runs of Tseng et al. [1] (see their Figure 5): one can observe that RTN converges faster in the beginning, but is overtaken by SR during the later stages of the training. As such, we claim that PMA effects will persist at that scale.
>
> > How would Equation (1) break if quantization error interacts non-linearly with model width or depth?
>
> Equation (1) models the loss for a particular training setup, determined, among everything else, by model hyper-parameter (depth, width, LR) scaling with parameter-count. In our experiments, we scaled model depth and width proportionally, leading to depth \~ width \~ N^(⅓). One can notice, that if quantization had separate multiplicative effects on depth and width, it could still be represented as a multiplicative factor on N. To separate these effects, one would need to construct a scaling law that would decouple depth from width and accurately model the effect of quantization as a function of these two parameters. That could possibly lead to an alternative depth-to-width strategy in the presence of quantization. That would constitute an interesting future research direction.
>
> > The implementation of Quartet is specialized for the MXFP4 format.
>
> We chose the MXFP4 format as it is the standard OCP Microscaling Format, with broader possible adoption, since it may also be supported by AMD devices.
> However, our kernel design is compatible with NVFP as well, and should perform similarly in terms of performance, and slightly better in terms of accuracy (since the scale quantization is more flexible in NVFP).
> We are working on an NVFP version of Quartet.
>
> Although NVFP4 could provide slightly higher accuracy due to smaller group sizes and finer scales, we expect it to perform similarly to MXFP4 due to similar  distributional properties of the underlying FP4 format that were shown to be connected to efficiencies in Table 2. Additionally, limited range of the FP8 shared exponents of NVFP4 could create problems for the backward pass with small gradient norm, requiring additional scaling techniques to prevent underflows.
>
> > Table 3 shows that the LSS-INT4 baseline becomes unstable…
>
> In our experiments, this baseline doesn’t simply diverge, but rather produces NaN iterations out of nowhere and without prior loss spikes. Its performance being close to Quartet while stable and the use of an unbiased backward pass indicate good alignment. We theorize that the lack of Hadamard preprocessing on the gradient outliers in the backward pass is what’s leading to overflows for LSS-INT4. Determining the exact cause would require extensive debugging of their implementation, which is unfortunately beyond the scope of our work.
>
> > Great speedup results in (Figure 3)...
>
> From our benchmarks, quantization-related operations constitute 20-40% of latency depending on the batch size and layer shape. We provide the breakdown for certain matrix shapes in “A.3 Performance breakdown” in Appendix.pdf in supplementary materials. Additionally, see response to reviewer **PgVh** for slightly adjusted performance measurements.
>
> > … providing a higher-level Triton implementation …
>
> We do provide a full reference Triton pseudo-quantization implementation in `./notebooks/benchmark_mxfp4.ipynb` in supplementary materials. In fact, we used it for some of the training runs because they were performed on Hopper-generation machines without hardware FP4 support.
>
> [1] https://arxiv.org/abs/2502.20586

---

> > ### Author Response · Authors · 2025-08-07
> > **Call for discussion**
> >
> > Dear reviewer **dMMD**.
> >
> > As the discussion period is drawing to a close, we would like you to let us know whether your concerns were addressed and if there are any further clarifications we can provide. Thank you for your time and feedback

---

### Decision · Program_Chairs · 2025-09-17

**Decision:**

Accept (poster)

**Comment:**

This paper introduces an algorithm called Quartet, which allows for FP4 training on the NVIDIA Blackwell architecture. All reviewers agree that this work makes solid contributions, and the AC recommends acceptance.